# Genetic Changes in Thyroid Cancers and the Importance of Their Preoperative Detection in Relation to the General Treatment and Determination of the Extent of Surgical Intervention—A Review

**DOI:** 10.3390/biomedicines10071515

**Published:** 2022-06-27

**Authors:** Jiri Hlozek, Barbora Pekova, Jan Rotnágl, Richard Holý, Jaromir Astl

**Affiliations:** 1Department of Otorhinolaryngology and Maxillofacial Surgery, Military University Hospital, 16902 Prague, Czech Republic; jan.rotnagl@uvn.cz (J.R.); richard.holy@uvn.cz (R.H.); jaromir.astl@seznam.cz (J.A.); 2Third Faculty of Medicine, Charles University, 10000 Prague, Czech Republic; 3Department of Molecular Endocrinology, Institute of Endocrinology, 11694 Prague, Czech Republic; bpekova@endo.cz

**Keywords:** thyroid carcinoma, molecular genetics, FNAC, fusion genes, mutations, surgical treatment, extent of surgery, neck dissection, prognosis

## Abstract

Carcinomas of the thyroid gland are some of the most common malignancies of the endocrine system. The causes of tumor transformation are genetic changes in genes encoding cell signaling pathways that lead to an imbalance between cell proliferation and apoptosis. Some mutations have been associated with increased tumor aggressiveness, metastatic lymph node spread, tendency to dedifferentiate, and/or reduced efficiency of radioiodine therapy. The main known genetic causes of thyroid cancer include point mutations in the *BRAF*, *RAS*, *TERT*, *RET*, and *TP53* genes and the fusion genes *RET*/*PTC*, *PAX8*/*PPAR-γ*, and *NTRK*. Molecular genetic testing of the fine needle aspiration cytology of the thyroid tissue in the preoperative period or of the removed thyroid tissue in the postoperative period is becoming more and more common in selected institutions. Positive detection of genetic changes, thus, becomes a diagnostic and prognostic factor and a factor that determines the extent of the surgical and nonsurgical treatment. The findings of genetic research on thyroid cancer are now beginning to be applied to clinical practice. In preoperative molecular diagnostics, the aggressiveness of cancers with the most frequently occurring mutations is correlated with the extent of the planned surgical treatment (radicality of surgery, neck dissection, etc.). However, clear algorithms are not established for the majority of genetic alterations. This review aims to provide a basic overview of the findings of the most commonly occurring gene mutations in thyroid cancer and to discuss the current recommendations on the extent of surgical and biological treatment concerning preoperatively detected genetic changes.

## 1. Introduction

Thyroid carcinomas (TC) represent approximately 1% of all malignancies, but they are also the most common endocrine malignancies with a frequency of more than 90% [1]. The most common TC, papillary carcinoma (PTC), followed by follicular (FTC), low-differentiated, and the rare but most aggressive anaplastic carcinoma (ATC), arises from the follicular cells of the thyroid gland [2]. Medullary carcinoma (MTC) originates from the parafollicular cells of the thyroid gland [2].

The known risk factors for the development of TC are ionizing radiation, iodine deficiency or excess, high levels of thyroid-stimulating hormone, and genetic predisposition [1,2].

Ultrasonography, fine needle aspiration cytology (FNAC), histopathological examination of the thyroid gland tissue, and laboratory tests with the possibility of measuring the tumor markers thyroglobulin and calcitonin are the basic diagnostic tools for the disease [2,3]. Molecular genetic testing of the FNAC sample is becoming more and more common in certain institutions [3]. Some mutations are specific for certain types of carcinomas and can therefore contribute to molecular diagnosis in the preoperative period, including refinement of the diagnosis in cytologically unclear findings (Bethesda categories III–V) [3]. Signs of increased tumor aggressiveness, metastatic lymph node involvement, and a tendency to dedifferentiate or reduced efficacy of radioiodine therapy have been described in association with some mutations [4,5,6]. However, for many mutations, the correlation between prognosis and the biological behavior of the tumor is so far unclear.

The American Thyroid Association (ATA), the European Thyroid Association (ETA) and the European Society for Medical Oncology propose the recommended treatments for thyroid disease [7,8,9]. Surgery, radioiodine therapy, radiotherapy, hormonal therapy, and, in recent years, expanding biological therapy related to the findings of molecular genetic analysis are the basis of therapy for thyroid malignancies and metastases to the affected lymph nodes [2,7,8,9]. However, the importance of preoperative detection of gene mutations associated with thyroid oncology in relation to the eventual escalation or de-escalation of surgical treatment is neither clearly addressed nor supported by sufficient scientific evidence in the current guidelines.

Our review aims to provide a basic overview of the most frequently occurring gene mutations in TC and to discuss the possibilities of nonsurgical and surgical treatment and its extent with regard to the risk of morbidity and prognosis of the disease in relation to the preoperative diagnosis of genetic changes.

## 2. Genetic Basis of Thyroid Tumors

The principle of tumor transformation and progression is the disruption of cell signal pathways regulating the balance between cell proliferation and apoptosis (Figure 1) [10]. Genetic changes occurring in genes encoding proteins of the MAPK (mitogen-activated protein kinase) signal pathway, which plays an important role in a wide range of cellular processes, such as regulation of gene expression, proliferation, differentiation, and programmed cell death, and in genes encoding the PI3K-AKT signal pathway, which plays a role in the regulation of glucose metabolism, survival, adhesion, and cell motility, have been described in the pathogenesis of thyroid cancer [10,11]. Two molecular mechanisms are mainly involved in thyroid cancer—point mutations and chromosomal rearrangements (fusion genes) [11]. In the case of point mutation, a single nucleotide is changed. In the case of chromosomal rearrangement, two different genes are fused [11]. Most of these genetic changes are nonhereditary—somatic (mutations arising directly in the thyroid tissue) [11]. Hereditary germline mutations are typical for familial forms of MTC and multiple endocrine neoplasia syndromes (MEN2A, MEN2B) [11]. Somatic mutations are analyzed from suspected thyroid tissue (FNAC, resected thyroid tumor tissue) [10]. Germline mutations are analyzed from peripheral blood collected from the patient and possibly his/her relatives [10].

Currently, point mutations in *BRAF*, *RAS*, *TERT*, *RET*, *TP53*, and the fusion genes *RET*/*PTC*, *PAX8*/*PPAR-γ*, *NTRK* are the main known genetic causes of TC [10,11]. The investigation of other causative gene mutations that are still unknown is the subject of intensive research [5,12,13].

### 2.1. BRAF

*BRAF* is a proto-oncogene encoding a cytoplasmic kinase that is involved in the MAPK pathway and is thus responsible for the regulation of cell proliferation, differentiation, and programmed cell death [11].

The 15th exon of the *BRAF* gene contains the most common PTC point mutation (valine to glutamate substitution at codon 600—*BRAF V600E*), causing permanent activation of the *BRAF* protein [11,13]. The mutated BRAF protein affects the expression of several genes, including reduced expression of the NIS (natriumiodide symporter) gene and genes for thyroglobulin and thyroperoxidase [14,15].

In association with this mutation, a worse prognosis and more frequent disease recurrence have been described, associated with higher tumor aggressiveness, extrathyroidal spread, local and distant lymph node metastases, and reduced effect of radioiodine therapy due to reduced iodine transport into the cell [4].

The occurrence of the *BRAF V600E* mutation is associated with an almost 100% risk of malignancy and occurs in approximately 30–70% of PTC and 30–40% of ATC [16]. According to the ETA’s recommendations, total thyroidectomy and consideration of region VI elective neck dissection is recommended in case of preoperative detection of *BRAF V600E* in nodules larger than 1 cm [8].

### 2.2. RAS

*RAS* genes encode a group of proteins that transmit a signal from the transmembrane tyrosine kinase receptor to the nucleus via the MAPK or PI3K-AKT pathway and are therefore important in cell growth and differentiation [5].

A point mutation in these genes can transform a proto-oncogene into an oncogene with subsequent stimulation of cell proliferation and inhibition of cell differentiation [5,6]. The most frequently detected mutations in the *HRAS*, *KRAS*, and *NRAS* genes can occur in benign and malignant thyroid tumors, and their significance is still not completely clear [6]. Mutations in *RAS* genes are considered to be an early transforming event and may predispose to progression to carcinoma in benign tumors [5,6,17].

The occurrence of *RAS* mutation is described in approximately 20% of follicular variants of PTC, 20–40% of FTA, 40–50% of FTC, and 20–40% of ATC [11,18]. After *BRAF* mutation, it is the most frequently detected genetic alteration in thyroid biopsies [19]. The risk of malignancy when *RAS* mutation is present in thyroid biopsies is reported differently for each gene: *HRAS* = 70.7%, *NRAS* = 63.4%, and *KRAS* = 33% [19].

According to the ETA’s recommendations, when pathogenic variants in *RAS* genes are detected, less radical surgery (e.g., hemithyroidectomy) is recommended, but other clinical and anamnestic data of the patient should be considered [8].

### 2.3. RET

The *RET* proto-oncogene encodes a transmembrane tyrosine kinase receptor and plays a key role in cell growth, differentiation, and survival [11]. Point mutations in the *RET* gene are typical for MTC [5]. Germline mutations are diagnosed in more than 95% of MEN2A and MEN2B patients, whereas the incidence of germline mutations in familiar MTC and somatic mutations in sporadic MTC is lower (50%) [20].

Genetic screening is recommended for relatives of patients with a detected germline mutation in the RET gene [12]. In case of an inherited *RET* mutation, there is a high risk of MTC, based on which a preventive total thyroidectomy is recommended [7,9].

Based on the described genotype–phenotype correlations, individual recommendations are established, especially regarding the timing of prophylactic total thyroidectomy in childhood to prevent the development of the disease [21].

### 2.4. RET/PTC

The chromosomal rearrangement *RET*/*PTC* is the most common mutation in TC in children and adolescents [13]. This fusion gene is more commonly associated with the classic papillary variant of TC and is associated with more aggressive tumor behavior and frequent metastatic dissemination [22]. Cases of carcinomas arising after radiation have been described in the past [2,22]. Detection of RET/PTC rearrangement is a strong indicator of PTC and may help in the FNAC molecular diagnosis, especially in cases of uncertain cytological findings. If *RET/PTC* rearrangement is detected, total thyroidectomy is recommended [8].

### 2.5. PAX8/PPAR-γ

*PAX8/PPAR-γ* is a fusion gene arising from a chromosomal rearrangement between the 2. and 3. chromosomes [11]. The *PAX8* gene is a transcription factor and is involved in the expression of genes for thyroglobulin, peroxidase, and NIS [11]. *PPAR-γ* is involved in lipid metabolism, cell differentiation, cell growth inhibition, and apoptosis [23]. This rearrangement occurs in approximately 1–5% of PTC, up to 60% of FTC, and 2–10% of FTA (however, the latter shows an immunohistochemical profile typical for carcinomas) [24].

In the case of preoperative FNAC findings of *PAX8/PPAR-γ*, the reported risk of malignancy is 84.6–95% [8,19]. The finding of a fusion gene should lead to a more thorough histopathological examination [19,24]. In case of *PAX8/PPAR-γ* fusion gene detection, total thyroidectomy is indicated, according to ETA recommendations [8].

### 2.6. TERT

The *TERT* gene encodes the catalytic subunit of the enzyme telomerase, which is responsible for telomere elongation during DNA replication [11]. Two major point mutations, *C228T* and *C250T*, have been described in association with this gene [25,26]. Cancer cells positive for one of these mutations express telomerase at an increased level, and thus maintain the length of chromosomal telomeres and can proliferate almost without any limitations [26].

*TERT* mutations are associated with a higher incidence of local or distant metastasis and tumor aggressiveness [25]. Mutations have been found in poorly differentiated thyroid carcinomas, ATC, and more aggressive forms of PTC [25,27]. Coexistence with the *BRAF V600E* mutation is associated with higher tumor aggressiveness in PTC than in *TERT* and *BRAF* mutations occurring alone [28].

According to ETA recommendations, in case of preoperative detection of *TERT* mutation in nodules larger than 1 cm, total thyroidectomy and consideration of region VI elective neck dissection is recommended [8].

### 2.7. NTRK Fusion Genes

*NTRK* fusion genes are described in 5–10% of PTC, and in pediatric and adolescent patients, the prevalence is around 16% [29]. The finding of an *NTRK* fusion gene in a thyroid sample is associated with a 100% risk of malignancy [12,29]. NTRK fusion-positive carcinomas usually show the follicular arrangement and are associated with the occurrence of chronic lymphocytic thyroiditis and frequent lymph node metastasis [5].

### 2.8. TP53

*TP53* is a tumor suppressor gene that regulates cell growth—it blocks the cell cycle in the G1 phase and initiates apoptosis in case of nonrepairable DNA [11]. High expression and mutations of *TP53* are detected in more than 75% of invasive and undifferentiated carcinomas [12]. The presence of *TP53* mutations in differentiated carcinomas has been described as a possible sign of subsequent dedifferentiation in ATC [5].

### 2.9. miRNA

*miRNAs* are short single-stranded noncoding ribonucleic acids (RNAs) that are involved in gene regulation [30]. More than one-third of structural genes in the human body are regulated by *miRNAs*; thus, dysregulation of *miRNAs* can lead to many specific cancer conditions [31]. Abnormal expressions of numerous *miRNAs* have been described in TC [31,32]. A recent study found that different *miRNA* profiles are associated with unique histological variants and *BRAF* mutations in PTC, but their influence on thyroid oncology is a relatively new finding and is still under scientific research, and further studies focusing on the relationship between *miRNA* function and thyroid cancers are needed [32].

## 3. Thyroid Cancer Therapy in Relation to the Findings of Molecular Genetic Analysis

The discovery of new molecular genetic markers of carcinomas of the thyroid gland in relation to the clinical course of the disease, preoperative and perioperative findings, surgical treatment outcomes, and risk of recurrence and persistence of malignant disease leads to new insights. Thanks to genetic analysis and the experiences gained, it is now possible to personalize the treatment of TC in regard to the indication and determination of the extent of surgical treatment and the use of systemic targeted therapy (tyrosine kinase inhibitors—TKIs; monoclonal antibodies—mAbs).

### 3.1. Surgical Treatment

Positive detection of mutations (*BRAF*, *RAS* in MTC, *RET*, *RET*/*PTC*, *PAX8*-*PPAR-γ*, *TERT*, *NTRK*) is an indication for total thyroidectomy and thus prevents the removal of the thyroid gland in two stages and the associated risks (general anesthesia, timing of oncological treatment) [8]. However, preoperative detection of genetic changes may also lead to an extension of surgical treatment to the lymph node catchment area [7,8,9]. Because of the relatively high incidence of occult metastases in PTC (20–40%) and the associated higher risk of disease recurrence and mortality, the indication of elective block neck dissection of region VI in cN0 patients is still a matter of debate [33,34,35,36,37,38,39,40].

The detection of *BRAF* mutation as a possible predictive factor of occult metastasis could lead to the indication of elective neck dissection of region VI, which in the case of positive metastatic lymph nodes would lead to a reduction of recurrence and refinement of the extent of the disease (staging—conversion from cN0 to pN1a) [15,41,42,43,44,45,46]. However, the change in surgical treatment intensification (neck dissection routinely indicated) based on the identification of mutations in the preoperative period in cN0 PTC is still controversial because of the potential increased risk of surgical complications (recurrent laryngeal nerve injury, hypoparathyroidism) and its relation to the overall treatment benefit (persistence, recurrence of disease) [33,34,35,41,42].

Although the ATA mentions elective neck dissection for T3 and T4 TC and the ETA for *BRAF*-positive TC greater than 1 cm in size, they still recommend “only” consideration of elective surgery [7,8]. The subject of current and future works is prospective long-term studies to assess the potential role of *BRAF* and other thyroid mutations that will lead to clear and more precise recommendations regarding the extent of surgical treatment with respect to its risks and benefits [2,3,5,12].

In the following section, we present the results of recently published studies that address the general risks (temporary/permanent recurrent laryngeal nerve involvement, temporary/permanent hypoparathyroidism) and benefits (reduced incidence of persistence/recurrence of disease, reduced incidence of lymph node metastasis) of elective neck dissection of region VI in patients with thyroid carcinomas (especially PTC) [33,34,35,36,37,40]. We also present the results of studies focusing on performing neck dissections of region VI specifically in patients with *BRAF V600E*-positive mutation, where *BRAF V600E* mutation in PTC is correlated with a higher risk of occult metastasis, and it is discussed whether it should be a determining factor for indicating elective neck dissection of region VI under the assumption of improved survival and reduced risk of recurrence of the primary disease [15,41,42,43,44,45,46].

The results of studies comparing the benefits of elective neck dissection and the increased risk of postoperative complications in patients with PTC are different, with many authors pointing out the inappropriateness of routinely performed elective neck dissection [33,34,35] in contrast to authors who recommend extended surgical treatment [36,37,38,39,40]. Publications suggesting that routine elective neck dissection is not recommended show a statistically significantly higher incidence of postoperative surgical complications in terms of temporary damage of the recurrent laryngeal nerve, temporary hypoparathyroidism, and permanent hypoparathyroidism, while finding no statistically significant difference in the overall survival and recurrence rate (including metastasis) of the disease in the group of patients operated by total thyroidectomy and neck dissection of region VI compared with the group of patients operated by total thyroidectomy only [33,34,35]. However, ipsilateral neck dissection of region VI could be an interesting option due to the lower rate of hypocalcemia [34]. On the other hand, authors of studies recommending total thyroidectomy concurrently with routine elective neck dissection point to statistically significant reduced risk of locoregional recurrence, reduced incidence of metastases to central neck lymph nodes, and reduced need for reoperations (these may lead to higher risk of surgical complications than first-time surgery) [36,37,40]. Routinely performed elective neck dissection may lead to the refinement of the pathological staging (pTNM classification) of the disease, however, further large prospective randomized trials are needed to properly assess the risk/benefit for cN0 PTC patients [35,37].

Due to the increased risk of temporary and permanent hypoparathyroidism in routinely performed elective neck dissection of the region VI and the different results of studies investigating the relationship of elective neck dissection with disease recurrence and survival in cN0 patients, risk factors related to metastatic spread to the central cervical lymph nodes have recently been studied. In the future, knowledge of these important risk factors should lead to the development of a diagnostic–therapeutic approach in order to predict the occurrence of metastases in cervical lymph nodes (not only central ones) and to indicate an adequate extent of surgical treatment in the preoperative period with regard to the high benefit and low risk of postoperative complications [15,41,42,43,44,45,46]. In recent studies, the most frequently mentioned predictive molecular factor for the occurrence of occult metastases to central cervical lymph nodes in cNO patients is the *BRAF V600E* mutation [15,38,39,43,44,45,46]. In large meta-analyses, male gender, age, tumor size > 10 mm, multifocality, tumor bilaterality, thyroid capsule affected by tumor, angiolymphatic invasion, and high histological risk have been correlated as additional predictive clinicopathological factors [42,44,45,46]. The presence of risk factors and especially their combination can be successfully used to differentiate cN0 patients with or without central neck lymph node metastases, and in particular, the simultaneous occurrence of *BRAF V600E* mutation with other risk factors can be a decisive factor for the extent of surgical treatment [15,44]. At the same time, *BRAF V600E*-positive patients who underwent elective neck dissection of region VI showed statistically significant improvement in disease survival without recurrence [45]. On the other hand, there are also published papers that do not consider *BRAF V600E* mutation as a predictive factor for the occurrence of lymph node metastases and that describe the same treatment outcome in cN0 patients with PTC *BRAF V600E*-positive mutation regardless of the extent of surgery (total thyroidectomy only vs. total thyroidectomy and neck dissection of region VI) and do not recommend elective neck dissection based on *BRAF V600E* detection, despite the fact that patients who have undergone neck dissection undergo repeated radioiodine treatment at a lower rate [41,42].

The role of elective neck dissection in the treatment of PTC without clinically present lymph node metastases remains controversial, especially regarding indications, approach, and surgical extension of treatment [32,34,35,36,37,40]. Although morbidity and survival rates appear to be similar to those reported for total thyroidectomy alone, the impact of neck dissection VI on local recurrence and long-term survival is still under investigation [36,37,40,45]. Currently, most publications recommend selective indications for elective neck dissection in patients diagnosed with risk factors or combinations of risk factors (most frequently mentioned: *BRAF* mutations, tumor size, multifocality, extracapsular spread) rather than routinely indicated lymph node procedures [15,44,45,46].

The current recommendations for surgical treatment in relation to preoperatively diagnosed genetic changes in TC are summarized in Table 1.

### 3.2. Targeted Therapy

TKIs are cytostatics that block the growth of cancer cells, so their use can stabilize or improve cancer, but not cure it completely [47]. Because of the number of alternative signal pathways that are able to be activated when one “targeted” pathway is blocked, targeted therapy may not be fully effective [47,48].

Side effects, such as diarrhea, vomiting, fatigue, arterial hypertension, hepatotoxicity, and rash, have been described during the use of TKIs. TKI therapy is reserved for patients with rapid progression of radioiodine-refractory thyroid cancer and is therefore indicated as an additional treatment to the primary therapy for thyroid cancer (surgery, radioiodine therapy, radiotherapy) [20,47,48,49,50]. According to the mechanism of effect, TKIs can be divided into multikinase inhibitors (inhibiting multiple intracellular and cell surface kinases) and mutation-specific inhibitors (selectively inhibiting the tyrosine kinase or kinases that are involved in the aberrant signaling way). The most frequently used are:

#### 3.2.1. Multikinase Inhibitors

Sorafenib—indicated for radioiodine refractory differentiated TC (target: VEGF1–3, PDGFR, RET, BRAF, c-kit). The drug has also been successful in patients with metastatic PTC, FTC, ATC, and MTC [51].

Lenvatinib—indicated for the treatment of patients with progressive, locally advanced, or metastatic differentiated radioiodine refractory thyroid cancer (target: VEGFR1–3, PDGFR, RET, BRAF, c-kit) [52].

Vandetanib—indicated for the treatment of aggressive and symptomatic MTC in patients with unresectable, locally advanced, or metastatic disease (target: VEGFR2/3, EGFR, RET). In metastatic MTC, it is a drug with low toxicity for surrounding tissues and results in the reduction of the growth of the primary tumor and its metastases [53].

Cabozantinib—indicated for the treatment of adult patients with progressive, inoperable locally advanced, and/or metastatic medullary thyroid carcinoma (target: VEGFR2, RET, MET, FLT3, c-kit) [53].

#### 3.2.2. Specific Inhibitors

Larotrectinib—NTRK inhibitor indicated in monotherapy for the treatment of adult and pediatric patients with solid tumors expressing the *NTRK* fusion gene who have locally advanced, metastatic disease and/or for whom surgical resection would likely result in significant morbidity [54].

Selpercatinib and pralsetinib—indicated for advanced or metastatic thyroid cancer with the positivity of a *RET* point mutation or the *RET* fusion gene. Selpercatinib is indicated in monotherapy for the treatment of adults and adolescents with advanced medullary thyroid cancer with *RET* mutation who require systemic therapy following prior treatment with cabozantinib and/or vandetanib [55].

Dabrafenib and trametinib—selective BRAF and MEK1/2 inhibitors indicated for patients with anaplastic thyroid cancer in whom the *BRAF V600E* mutation is present. This combination appears to be a promising treatment for its overall response rate and duration of response with controllable toxicity [56].

## 4. Conclusions

Nowadays, the findings of genetic research on thyroid cancer are beginning to be applied to clinical practice [12]. In preoperative molecular FNAC diagnostics, the aggressiveness of cancers with the most frequently occurring mutations (*BRAF*, *RAS*, *RET*/*PTC*, *TERT*, *PAX8*/*PPAR-γ*, *RET proto-oncogene*) is correlated with the extent of the planned surgical treatment [32,33,34,35,36,37,38,39,40]. However, there are no established guidelines for most genetic alterations, and their use in the preoperative and postoperative periods is still debated.

Total thyroidectomy remains the primary procedure for thyroid carcinomas [7,8,9]. Currently, the most studied mutations are *BRAF V600E*, which is 100% specific for malignancy, and germline mutations of the *RET proto-oncogene*, the detection of which is an indication for prophylactic surgical treatment [8,16,21]. Preoperative detection of mutations leads to: (1) more radical thyroid surgery (total thyroidectomy instead of hemithyroidectomy) and (2) consideration of region VI elective neck dissection in cN0 patients [7,8,9,12,15,43,44,45,46].

In the last decade, the nonsurgical treatment of advanced thyroid cancer has undergone great development. Currently, single-targeted (mutation specific) and multitargeted therapies can be used, particularly for aggressive, refractory, and metastatic thyroid carcinomas that do not respond to surgical and radioiodine treatment.

The concept of elective neck dissection in patients with cN0 PTC is still debated with regard to the risks and benefits of surgical treatment of the cervical lymph nodes. The subject of further research is the identification of risk factors (not only genetic) correlated with metastatic spread to the lymph nodes and their application in the preoperative period.

The objective of further research is to identify the genetic cause of different types of thyroid cancer and to understand the nature of the disease, its development, and prognosis. These findings may lead to personalized and targeted patient treatment and to an update of the recommended surgical management of thyroid cancer.

## Figures and Tables

**Figure 1 biomedicines-10-01515-f001:**
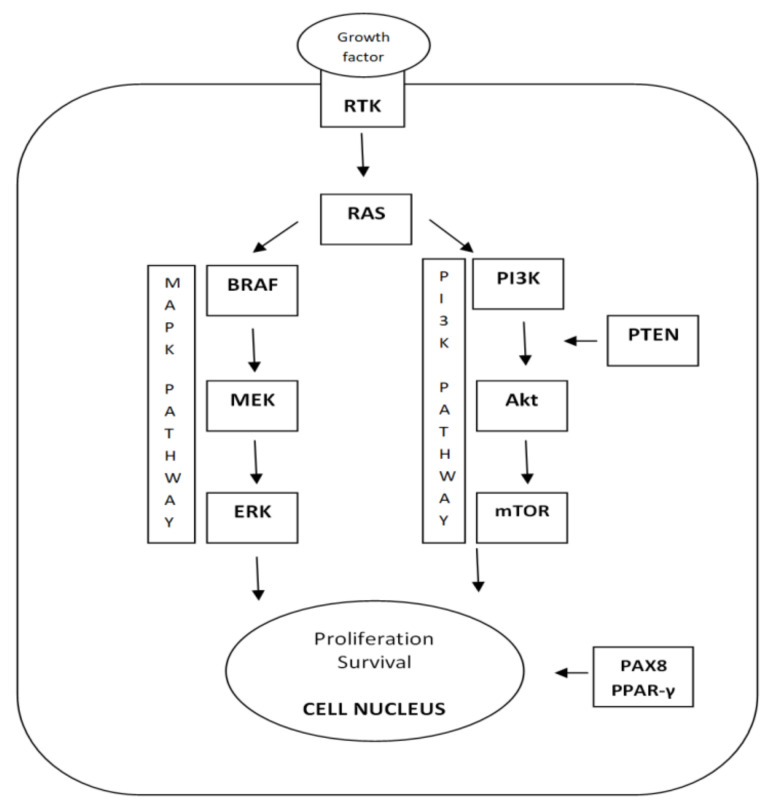
Simlified scheme of cell signaling pathways important in thyroid cancerogenesis (RTK—growth factor receptortyrosine kinase; RAS, BRAF, MEK, ERK, PI3K—signal molecules; Akt—protein kinase B; mTOR—mechanistic target of rapamycin; PTEN—phosphatase and tensin homolog; *PAX8/PPAR-γ*—fusion gene).

**Table 1 biomedicines-10-01515-t001:** The most common mutations and their detection in thyroid tumors (FTA—follicular adenoma, PTC—papillary carcinoma, FTC—follicular carcinoma, MTC—medullary carcinoma, ATC—anaplastic carcinoma, TT—total thyroidectomy, HT—hemithyroidectomy, ND—neck dissection).

	FTA	PTC	FTC	MTC	ATC	Recommended Extent of Surgery
** *BRAF* **	-	30–70%	-	-	25–35%	TT, consider elective ND
** *RAS* **	20–40%	10–20%	40–50%	*HRAS, KRAS* in 20–40% sporadic MTC	20–40%	HT; In suspected MTC: TT + ND
** *RET* **	-	-	-	95% familiar 50% sporadic	-	(Prophylactic) TT + ND
** *RET/PTC* **	-	20%	-	-	-	TT
** *PAX8* ** ** *PPAR-γ* **	Occasionally	1–5%	30–35%	-	-	TT
** *TERT* **	-	11%	17%	-	40%	TT, consider elective ND
** *NTRK* **	-	5–10%	-	-	-	TT
** *TP53* **	Occasionally	Occasionally	-	-	60–70%	-
** *PIK3CA* **	Occasionally	-	10–30%	-	25–45%	-
** *PTEN* **	Occasionally	-	8–10%	-	6%	-

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
