# Peer review of "Genetic Changes in Thyroid Cancers and the Importance of Their Preoperative Detection in Relation to the General Treatment and Determination of the Extent of Surgical Intervention—A Review"

_biomedicines, 2022, doi:10.3390/biomedicines10071515_

Round 1

Reviewer 1 Report

The article by Hložek J and colleagues entitled “GENETIC CHANGES IN THYROID CANCERS AND THE IM- 2 PORTANCE OF THEIR PREOPERATIVE DETECTION IN RE- 3 LATION TO THE GENERAL TREATMENT AND DETERMI- 4 NATION OF THE EXTENT OF SURGICAL INTERVENTION - 5 A REVIEW” is a very interesting manuscript. Overall, this review is well written and the authors have to address some minor concerns before it can be published in the journal “biomedicines”.

My minor concerns are:

- The authors should introduce abbreviations only if these abbreviations are used in the text.

- The abbreviation FNAC must be deleted in the abstract.

- Very often citations for statements are missing; the authors have to prove all statements with citations; e.g. lines 42, 49, 51, 63, 81, 82, 84, 85, 86, 87, 89, 95, 98, 115, 117, 122, 130, 131, 135, 136, 142, 145, 150, 151, 157, 161, 162, 166, 167, 174, 176, 181, 183, 187, 188, 189, 205, 207, 225, 227, 230, 231, 239, 249, 255, 269, 273, 282, 288, 294, 298, 302, 306, 313, 319, 326, 331, 333, 336, 339, 340, 351, 353, 357, 360, 364, 369, 380, 383, 385, 388, 390.

- If the authors wrote “…most publications….” more than one citation is necessary.

- The authors have to be more precise – they must give more details about “…various mutagens…” as well as “…other causative gene mutations….”.

- Figure 1 must be modified and the Figure legend should be presented below the Figure.

- Under point 2.9 the authors have to cite more publications dealing with microRNAs and TC, e.g. doi: 10.3389/fendo.2022.834075. eCollection 2022.

- It is neither nice nor necessary to read “Dobrinja C. et al..,”. ….” or “…. Unlu M.T. et al.  . ….” etc. if somebody is interested in the name of the first author (s)he will find this information in the Reference list. The authors have to rephrase all these parts.

- The authors have to delete lines 370 – 377. This part is not of interest for the reader nor important for this review.

Author Response

Dear Mr. Reviewer

Thank you for your comments and suggestions.

Based on your comments, I have made the requested changes and modifications (described in more detail below).

In addition to the requested changes, I have added minor modifications:

  • On line 10, I've edited the title "Third“ .. Faculty of medicine....
  • For the sake of clarity, I have divided the section "3. Therapy of thyroid cancer in relation to the results of molecular genetic analysis" into surgical and non-surgical treatment.

  • In the conclusion I mentioned the possibility of non-surgical treatment, which I discuss in more detail in section "3. Thyroid cancer therapy in relation to the findings of molecular genetic analysis".

  • Changes to the manuscript made outside of your comments were made based on the requests of reviewer 2.

Checking and correction of the English language was carried out by a native speaker.

Sincerely, JiÅ™í Hložek

----------------------------------------------------------------------------------------------------------------------------------------

- The authors should introduce abbreviations only if these abbreviations are used in the text.

abbreviation TG removed (line 42), abbreviation TSH removed (line 45), abbreviation ESMO removed (line 59), abbreviation FMTC removed (line 133)

- The abbreviation FNAC must be deleted in the abstract.

            corrected

- Very often citations for statements are missing; the authors have to prove all statements with citations; e.g. lines 42, 49, 51, 63, 81, 82, 84, 85, 86, 87, 89, 95, 98, 115, 117, 122, 130, 131, 135, 136, 142, 145, 150, 151, 157, 161, 162, 166, 167, 174, 176, 181, 183, 187, 188, 189, 205, 207, 225, 227, 230, 231, 239, 249, 255, 269, 273, 282, 288, 294, 298, 302, 306, 313, 319, 326, 331, 333, 336, 339, 340, 351, 353, 357, 360, 364, 369, 380, 383, 385, 388, 390.

            corrected

- If the authors wrote “…most publications….” more than one citation is necessary.

            corrected

- The authors have to be more precise – they must give more details about “…various mutagens…” as well as “…other causative gene mutations….”.

„various mutagens“ removed ... (the main mutagen ionising radiation and other factors leading to an increased risk of thyroid cancer have already been mentioned in the same sentence) line 45

„other causative gene mutations„ – corrected .. (meant to explore new, previously unknown mutations) line 90

- Figure 1 must be modified and the Figure legend should be presented below the Figure.

Corrected - if you don't mind, I left the Figure legent above the Figure directly after the image title (as in Picture 1)

- Under point 2.9 the authors have to cite more publications dealing with microRNAs and TC, e.g. doi: 10.3389/fendo.2022.834075. eCollection 2022.

            corrected

- It is neither nice nor necessary to read “Dobrinja C. et al..,”. ….” or “…. Unlu M.T. et al.  . ….” etc. if somebody is interested in the name of the first author (s)he will find this information in the Reference list. The authors have to rephrase all these parts.

based on your comment and reviewer's No.2  comment, this section has been shortened and rewritten to give an overall view of surgical treatment and neck dissections instead of describing individual publications

- The authors have to delete lines 370 – 377. This part is not of interest for the reader nor important for this review.

            corrected

Reviewer 2 Report

Hlozek and colleagues reviewed the genetic alterations in thyroid cancer regarding the molecular diagnosis and the impact on surgical treatment. I have a few comments:

(1) Genetic alterations generally include point mutations, gene fusions, and somatic copy number alteration. Some descriptions are not very clear, e.g. for NTRK fusions, Line 176 carcinomas positive for this 'mutation'
(2) Line 230-336: describing the results of each study rather than giving a whole picture. This is quite fragmented.
(3) Line 348-369: targeted therapy should be grouped into two classes: specific ones (e.g. cabozantinib, selpercatinib, larotrectinib, and dabrafenib+trametinib) and general ones (irrelevant to underlying genetic backgrounds, e.g. sorafenib and lenvatinib).
(4) Figure 1 is poorly presented.
(5) Avoid unnecessary, uncommon abbreviations, e.g. total thyroidectomy (TTE), hemithyroidectomy (HTE), neck dissection (ND). Abbreviations break fluidity and make reading the article very cumbersome.
(6) 2.5. PAX8/PPARy should be PPAR-γ (gamma).
(7) 2.9. MiRNA, spelled out as 'MicroRNA' or 'miRNA'
(8) Line 479:  Int J Surg

Author Response

Dear Mr. Reviewer

Thank you for your comments and suggestions.

Based on your comments, I have made the requested changes and modifications (described in more detail below).

In addition to the requested changes, I have added minor modifications:

  • On line 10, I've edited the title "Third“ .. Faculty of medicine....
  • For the sake of clarity, I have divided the section "3. Therapy of thyroid cancer in relation to the results of molecular genetic analysis" into surgical and non-surgical treatment

  • In the conclusion I mentioned the possibility of non-surgical treatment, which I discuss in more detail in section "3. Thyroid cancer therapy in relation to the findings of molecular genetic analysis"

  • Changes to the manuscript made outside of your comments were made based on the requests of reviewer 1.

Checking and correction of the English language was carried out by a native speaker.

Sincerely, JiÅ™í Hložek

----------------------------------------------------------------------------------------------------------------------------------------

(1) Genetic alterations generally include point mutations, gene fusions, and somatic copy number alteration. Some descriptions are not very clear, e.g. for NTRK fusions, Line 176 carcinomas positive for this 'mutation'

            corrected

(2) Line 230-336: describing the results of each study rather than giving a whole picture. This is quite fragmented.

Corrected. Based on your comment and reviewer's No.1  comment, this section has been shortened and rewritten to give an overall view of surgical treatment and neck dissections instead of describing individual publications

(3) Line 348-369: targeted therapy should be grouped into two classes: specific ones (e.g. cabozantinib, selpercatinib, larotrectinib, and dabrafenib+trametinib) and general ones (irrelevant to underlying genetic backgrounds, e.g. sorafenib and lenvatinib).

Corrected. With permission, I have divided TKIs into two groups: 1) multikinase inhibitors 2) specific inhibitors ...with reference to articles e.g. Targeted Therapy for Advanced Thyroid Cancer: Kinase Inhibitors and Beyond, doi: 10.1210/er.2019-00007

(4) Figure 1 is poorly presented.

corrected

(5) Avoid unnecessary, uncommon abbreviations, e.g. total thyroidectomy (TTE), hemithyroidectomy (HTE), neck dissection (ND). Abbreviations break fluidity and make reading the article very cumbersome.

Corrected. With permission, I used only the abbreviations TT - total thyroidectomy, HT - hemityhoridectomy, ND - neck dissection for clarity / simlicity in Table 1.

(6) 2.5. PAX8/PPARy should be PPAR-γ (gamma).

corrected

(7) 2.9. MiRNA, spelled out as 'MicroRNA' or 'miRNA'

corrected

(8) Line 479:  Int J Surg

            corrected

Round 2

Reviewer 2 Report

The authors have addressed all my concerns.